# When to Perform a Colonoscopy in Diverticular Disease and Why: A Personalized Approach

**DOI:** 10.3390/jpm12101713

**Published:** 2022-10-14

**Authors:** Antonio Tursi, Valerio Papa, Loris Riccardo Lopetuso, Lorenzo Maria Vetrone, Antonio Gasbarrini, Alfredo Papa

**Affiliations:** 1Territorial Gastroenterology Service, ASL BAT, 70031 Andria, Italy; 2Department of Translational Medicine and Surgery, School of Medicine, Catholic University, 00168 Rome, Italy; 3Digestive Surgery, Fondazione Policlinico Universitario “A. Gemelli”, Istituto di Ricovero e Cura a Carattere Scientifico, 00168 Rome, Italy; 4Gastroenterology Department, Centro Malattie Apparato Digerente, Center for Diagnosis and Treatment of Digestive Diseases, Fondazione Policlinico Gemelli, Istituto di Ricovero e Cura a Carattere Scientifico, 00168 Rome, Italy; 5Department of Medicine and Ageing Sciences, “G. d’Annunzio” University of Chieti-Pescara, 66100 Chieti, Italy; 6Center for Advanced Studies and Technology (CAST), “G. d’Annunzio” University of Chieti-Pescara, 66100 Chieti, Italy

**Keywords:** diverticular disease, colonoscopy, diverticular bleeding, segmental colitis associated with diverticulosis, DICA classification, acute diverticulitis

## Abstract

Colonoscopy is a crucial diagnostic tool in managing diverticular disease (DD). Diverticulosis can often be an unexpected diagnosis when colonoscopy is performed in asymptomatic subjects, generally for colorectal cancer screening, or it could reveal an endoscopic picture compatible with DD, including acute diverticulitis, in patients suffering from abdominal pain or rectal bleeding. However, alongside its role in the differential diagnosis of colonic diseases, particularly with colon cancer after an episode of acute diverticulitis or segmental colitis associated with diverticulosis, the most promising use of colonoscopy in patients with DD is represented by its prognostic role when the DICA (Diverticular Inflammation and Complication Assessment) classification is applied. Finally, colonoscopy plays a crucial role in managing diverticular bleeding, and it could sometimes be used to resolve other complications, particularly as a bridge to surgery. This article aims to summarize “when” to safely perform a colonoscopy in the different DD settings and “why”.

## 1. Introduction

The presence of sac-like protrusions that occur when colonic layers are herniated through defects in the muscle layer of the colon wall is called diverticulosis. Both the left and right colon may harbor diverticula but with different characteristics.

In the left-sided diverticulosis, we have herniation of mucosa and submucosa (“pseudo-diverticula”), while in the right-sided diverticulosis, we have herniation through all colonic layers (“real diverticula”) [1]. In westernized countries, the distribution of diverticula is predominantly left-sided; on the contrary, in Asian countries, they affect the right side of the colon. 

For many years, it has been thought that diverticulosis exclusively affects the westernized world, mainly due to a lack of fiber in the diet and following increased pressure in the colonic wall due to a reduction in the radius of the lumen [2,3]. Interestingly, the prevalence of diverticulosis seems to be growing worldwide in the last decade, both in the left and right colon [4]. Epidemiological studies have reported that the risk factors are similar between left- and right-sided diverticulosis [5]; consequently, the increased prevalence in Asia could be partially explained by adopting Western diets [4,6]. However, among the Asian population, the right-side predominance is not changed, which could be explained by the hypothesis that diet influences the incidence of diverticulosis but not the location. On the contrary, genetic factors seem to affect the site of diverticulosis [6]. Recently, a genome-wide association study in a Korean population identified several single-nucleotide polymorphisms associated with right-colonic diverticulosis [7].

In the westernized world, the prevalence of diverticulosis increases with age, and fewer than 20% of individuals younger than 40 years of age are noted to have diverticulosis on colonoscopy, compared with more than 60% of individuals older than 70 years [1]. However, the rate of complications such as acute diverticulitis (AD) or diverticular bleeding is relatively low. Recently, Sharara et al. showed that incidental diverticulosis was found in 224 of 823 (27.2%) (mean age 62.3 ± 8.2 years) of patients who underwent colonoscopy for colorectal cancer screening [8]. Over a mean follow-up of 7.0 ± 1.7 years, a diverticular complication occurred in 6 out of 144 patients (4.2%) (four cases of AD, one probable case of diverticular bleeding, and one case of AD and diverticular bleeding), with an incidence rate of 5.9 per 1000 patient-years [8]. These data are similar to those reported by a Veterans Affairs Greater Los Angeles Healthcare System study, in which AD occurred in only 4% of patients with diverticulosis [9], contradicting the common thought that diverticulosis has a high rate of progression. Diverticulosis of the colon is often an incidental finding and does not generally affect the safety or accuracy of colonoscopy. Sometimes, the presence of features of the colon, such as the presence of severe sigmoid diverticulosis or fixed angulation of the colon, and potential confusion as to the location of the true lumen when multiple large diverticular orifices are detected, increase the risk of colonic perforation [10,11,12]. However, colonoscopy represents the most important diagnostic and therapeutic tool in the hands of clinicians and plays a crucial role in different clinical settings of DD [13]. In this narrative review, we evaluate the clinical scenarios of DD wherein a colonoscopy should be performed. For this purpose, a literature search was conducted using Pubmed until August 2022. Original articles and reviews were identified using the keywords: “diverticulosis”, “diverticular disease”, “acute diverticulitis”, “diverticular bleeding”, “complicated diverticulitis”, and “segmental colitis associated with diverticulosis” matched with each of the following keywords: “colonoscopy”, “computed tomography colonography”, “endoscopic hemostasis”, “endoscopic dilation”. Additional articles were identified by reviewing the reference lists of selected pertinent articles.

## 2. Differential Diagnoses among Different Colonic Diseases Harboring Diverticula

Diverticulosis may not be the only disease that occurs in a colon harboring diverticula, and consequently, it may also be detected in patients suffering from other colonic disorders [14].

Segmental colitis associated with diverticulosis (SCAD) is a chronic inflammatory process in the colon harboring diverticula. Generally located in the sigmoid colon, this disease is relatively rare, with a prevalence of 0.25–1.4% in the general population and 1.15–11.4% amongst patients having diverticulosis [15,16,17]. It mainly affects older males (the mean age at diagnosis is 60 years), and rectal bleeding is the main symptom [18]. The pathogenesis is unknown but likely multifactorial, including genetic susceptibility, microbiome imbalance, local ischemia, mucosal prolapse, and more [17,18]. The endoscopic features of SCAD can be subdivided into four subtypes, which also reflect different histological characteristics [16,19]: type A (or “crescentic fold disease”) is characterized by red patches involving colonic folds and diverticular sparing, with neutrophil and lymphocyte infiltrates limited to crypt epithelium; type B (or “mild-to-moderate ulcerative colitis-like”) shows endoscopic and histological features resembling mild to moderate ulcerative colitis (UC), with erosions and hyperemic areas involving the colonic folds and the interdiverticular mucosa (Figure 1), as well as crypt distortion, cryptitis, and crypt abscesses; type C (or “mild to moderate Crohn’s disease-like”) is characterized by Crohn’s disease (CD)-like changes, with isolated aphthous ulcers and transmural inflammatory changes; type D (or “moderate to severe ulcerative colitis-like”) shows the same endoscopic and histological features as type B, but is more severe. Although there are histological similarities between SCAD and inflammatory bowel disease (IBD), endoscopic examinations can help to propose a correct differential diagnosis: (a) in SCAD, the inflammatory process involves the inter-diverticular mucosa in the colonic area presenting diverticulosis, and it is therefore located mainly in the sigmoid colon; (b) in SCAD, the rectum and proximal colon are endoscopically and histologically normal; (c) in UC, the rectum is nearly always affected, and CD may patchily affect the colon and other gastrointestinal districts. SCAD generally shows a benign course [20], rarely requiring an aggressive approach to obtain remission [21].

## 3. Colonoscopy Following Acute Diverticulitis

The most severe manifestation of DD is AD and its complications. AD diagnosis is based on the triad of left lower quadrant pain–fever–leukocytosis, even if characteristic CT scan findings seem more critical [1]. There is disagreement on the indications for colonoscopy in AD, while there is no doubt that colonoscopy should be performed in most patients after the resolution of AD [22]. Gastroenterologists should avoid colonoscopy in patients suspected of AD because air inflation and instrumental manipulation are considered risky for intestinal perforation. However, when imaging studies are equivocal, a colonoscopy may be required to correctly differentiate AD from any other segmental colonic abnormality that could be associated with diverticulosis [23]. In this situation, a gentle colonoscopy with minimal air inflation can be carried out safely. If AD is confirmed, the advice is to terminate the procedure at that point. Expert opinion favors colonoscopy when the acute process has been resolved, usually after approximately six weeks, to avoid the potential risk of converting a sealed perforation into a free perforation [24]. This approach is strongly advised not only to prevent a colorectal cancer misdiagnosis, but also because the persistence of endoscopic signs of inflammation may be a risk factor for AD recurrence [25]. Computed tomography colonography (CTC) is the diagnostic modality that can be used in selected cases (for example, elderly or frail patients, patients with multiple morbidities or unable to complete colonoscopy in the past, or who explicitly refuse to undergo colonoscopy) for the follow-up of AD [26,27]. CTC has the potential to obtain comprehensive endoluminal and extraluminal evaluation thanks to 2D and 3D reconstruction imaging [28]. In addition, it is less painful and unpleasant than a standard colonoscopy because of reduced bowel preparation and colonic distension [29]. However, the CTC does not allow the execution of biopsies or the removal of polypoid lesions and requires adequate expertise. On the contrary, barium enema (BE) should no longer find a place in the diagnostic work-up of DD because of poorer patient compliance, higher risk of complications, and, finally, radiation exposure [29].

### 3.1. Early Colonoscopy Following Acute Diverticulitis

A colonoscopy may be undertaken earlier (after seven days) after diagnosing AD if the patient’s symptoms do not improve (Figure 2). Lahat et al. showed the feasibility and safety of early colonoscopy during hospitalization in patients with AD and without pericolic air on a CT scan. The authors found this timing to be as safe as late colonoscopy, as recommended by current practice, and improved patient compliance (93.3% of the hospitalized patients underwent early colonoscopy vs. 75.6% of the outpatient group who had undergone late colonoscopy; *p* = 0.03) [30]. Furthermore, the same group more recently found that early colonoscopy for persistent symptoms after AD can help detect ongoing inflammatory endoscopic and histological findings that may guide the therapeutic choices in these patients [31].

### 3.2. Late Colonoscopy Following Acute Diverticulitis to Rule Out Colorectal Cancer

A late colonoscopy (≥6 weeks) following an AD episode is advised to rule out colorectal cancer. This advice arises from the data of several studies. Older studies searching for associations between DD and colorectal cancer have described a clear overall association suggesting that long-lasting diverticular inflammation can lead to cancer. However, the issue is still debated because these studies have usually been retrospective, case–control, cohort, or cross-sectional, and the number of patients included was mainly low (<1000 patients enrolled). The first study by Morini et al. showed an association between diverticulosis and colorectal adenomas [32]. The mechanisms of this link (if they existed) were usually speculated to be a consequence of long-standing chronic inflammation, alterations in the extracellular matrix in the involved colon segments promoting carcinogenesis, and increased cell proliferation with increased numbers of aberrant crypt foci [32,33]. However, all those hypotheses lack substantial evidence. The most recent studies involved larger cohorts of patients. The first significant nationwide case–control study involving 41,037 patients was done in Sweden by Granlund et al. [34]. This study matched each case of colorectal cancer identified by the Swedish Cancer Registry with two controls without cancer (82,074 patients). Cases and controls were then searched for episodes of hospitalizations with the diagnosis of DD, and odds ratios (ORs) for receiving the diagnosis of colon cancer were calculated after hospital discharge for DD. Finally, cancer mortality was calculated according to the presence or not of DD. The results of this study were significant. First, the OR for receiving the diagnosis of colon cancer was extremely high (31.49; 95% CI, 19.00–52.21) within six months following hospitalization for DD, but the risk of a diagnosis of colon cancer was not increased at all through a 12-month follow-up after hospitalization. Second, there was no difference in mortality between patients with and without DD. The authors concluded by giving strong recommendations that patients with DD should have a high-quality diagnostic work-up within the first 12 months following an initial episode of DD. In another large study from Taiwan, the authors retrieved a cohort of 41,359 patients with DD (28,909 with diverticulitis and 12,450 with diverticulosis) from the National Health Research Institute database, matched with four controls (165,436 individuals without DD) [35]. In the initial analysis, the colorectal cancer risk was significantly increased (adjusted Hazard Ratio (HR) = 4.54; 95% CI, 4.19–4.91) [28]. However, after excluding the first 12 months of follow-ups, the adjusted HR was not increased at all (HR = 0.98; 95% CI, 0.85–1.13), neither for patients with diverticulosis nor for patients with diverticulitis [35]. The authors concluded similarly to Granlund et al. [34]: colorectal cancer risk is not increased in patients with DD, and the increased risk within the first year after DD diagnosis instead suggests the misclassification and misdiagnosis of DD. Brar et al. analyzed the risk of colorectal cancer according to uncomplicated or complicated (namely, with pericolic or pelvic abscess at the time of presentation) diverticulitis [36]. The authors found that the advanced adenoma and colorectal cancer rate was significantly higher in complicated vs. uncomplicated diverticulitis (18.9% vs. 5.4% and 5.4% vs. 0%, respectively). On multivariate analysis, both age (OR = 1.04, 95% CI: 1.01–1.08) and the presence of intra-abdominal abscesses (OR = 4.15, 95% CI: 1.68–10.3) were determined to be independent risk factors. Finally, Lau et al. [37] found that the ORs for colonic malignancy in patients with complicated diverticulitis were 6.7 (95% CI: 2.4–18.7) in patients with abscesses, 4 (95% CI: 1.1–14.9) in patients with local perforation, and 18 (95% CI: 5.1–63.7) in patients with concomitant fistula when compared with uncomplicated diverticulitis. Considering these epidemiological data, two recent meta-analyses assessed the role of routine colonoscopy following an episode of AD [38,39]: both found that colonoscopy had a strong indication following a bout of complicated diverticulitis or in patients in whom a high-quality examination of the colon has not been recently performed [40].

## 4. A New Role for the Colonoscopy in Diverticular Disease: A Predictive Tool for Outcomes of the Disease

As already mentioned, different features can be detected through colonoscopy in patients with diverticula: non-inflamed diverticula, diverticulitis with or without complications, bleeding diverticula, and SCAD. In addition, we can find indirect signs of the previous AD, such as the rigidity of the colonic wall with scarce dilation of the intestinal lumen under inflation and the substenosis or stenosis of the colonic lumen (Figure 3). Due to the large number of colonoscopies currently performed worldwide in real life, these features are not so hard to detect [14,41]. However, neither a classification of the endoscopic appearance of the colon nor an analysis of the predicting value of these pictures was available until 2015. In 2015, an Italian group developed the first endoscopic classification of diverticulosis and DD, called DICA (Diverticular Inflammation and Complications Assessment) [42]. This classification aimed to give an objective and reproducible score to describe the endoscopic severity of DD. It is constituted of four main items (extension of diverticulosis, number of diverticula per district, the presence or not of inflammatory signs, and the presence or not of complications) and several sub-items (<15 to >15 diverticula per colonic district, edema/hyperemia, erosions, SCAD, stiffness or the luminal stenosis, bleeding and pus), each of which have a specific numerical score. The sum of the scores leads to three different DICA scores: DICA 1 (up to three points), DICA 2 (between four and seven points), and DICA 3 (over seven points) [42]. The DICA score can predict the outcomes of the disease in terms of AD occurrence/recurrence and surgery. Indeed, after the first promising results coming from a retrospective study [43], a recent large (more than 2000 patients enrolled) prospective study confirmed that DICA classification plays a significant role in predicting the outcome of the disease since the severity of the DICA score was strictly linked to the risk of having an AD occurrence/recurrence, or to a surgical procedure due to DD complications [44]. Moreover, the authors found that the clinical evolution of the DICA classification, the CODA (Combined Overview on Diverticular Assessment), combines the DICA score with two clinical parameters; namely, age at baseline and severity of abdominal pain at baseline, showed a more substantial 3-year cumulative probability of diverticulitis and surgery [44]. These results, together with an extensive validation process performed by experts [45,46] and in real life [47], confirm that DICA classification may significantly impact the therapeutic choice, and its development opens the way to further, more personalized therapeutic trials in these patients. However, every coin has its reverse; DICA classification works very well when applied to patients at the first DD diagnosis. In contrast, in patients with a prior diagnosis of DD and already treated, it works much less effectively [48] (Figure 4). Therefore, as for IBD patients [49], it is advisable to start the treatment for DD only after a correct diagnosis has been made to avoid misdiagnosis, misclassification, or unexpected adverse events.

## 5. Colonic Diverticular Bleeding

Colonic diverticular bleeding is the most common cause of lower gastrointestinal bleeding (LGIB) [1]. It affects 3% to 15% of patients with colonic diverticulosis, with a mortality rate of 2–3%. Several risk factors for diverticular bleeding have been identified in the last few years, such as alcohol consumption, smoking, and using non-steroidal anti-inflammatory drugs (NSAIDs)/antiplatelet drugs. Additional risk factors of diverticular re-bleeding are older age (>62 years), recurrence of diverticulitis, chronic renal failure, and peripheral vascular disorders [1].

The goal of colonoscopy in colonic diverticular bleeding is to identify the site of bleeding and perform hemostasis, which is indicated only for the diverticulum with stigmata of recent hemorrhage (SRH). Unfortunately, the detection rate of SRH is relatively low, and access to the bleeding site may not always be available. In patients with suspected colonic diverticular bleeding, colonoscopy is generally indicated in the following settings: (a) Electively, namely, when bleeding has stopped spontaneously (the most frequent condition occurring in 70–80% of cases). This approach is necessary to exclude other causes of LGIB, such as vascular ectasia or colonic neoplasia (Figure 5). (b) As a primary intervention in managing colonic diverticular bleeding (this condition occurs in 10% of cases). This condition requires an urgent colonoscopy to find signs of diverticular bleeding (active bleeding, visible vessel, or adherent clot). When approaching colonoscopy due to LGIB, significant attention must be paid to the timing of the colonoscopy and bowel preparation to improve the diagnostic and therapeutic yield. An urgent colonoscopy for acute LGIB must be performed within 24 h and is associated with a shorter length of hospital stay and lower hospitalization costs [50]. Unprepared colonoscopy must be avoided due to the low cecal intubation rate (55–70%) and high risk of bowel perforation [50,51,52]. (c) As direct imaging in patients with recurrent episodes of LGIB in which computerized tomography angiography was non-diagnostic (this condition occurs in 5% of cases). This approach could be underestimated since the re-bleeding rate in colonic diverticular bleeding ranges from 20 to 38% [53,54]. After the identification of the site of diverticular bleeding, the endoscopic therapy includes epinephrine injection, thermal coagulation, endoclip placement, endoscopic band ligation (EBL), snare ligation (EDSL), over-the-scope clip (OTSC), and hemostatic powder [55,56,57]. All these techniques seem to be effective in controlling both bleeding and re-bleeding. Endoscopic hemostasis is generally safe and is rarely complicated by perforation or acute diverticulitis. Ligation therapy may induce diverticulitis in less than 1%, while EBL may cause delayed colonic perforation 4–5 days after the procedure [58].

## 6. Other Indications for Operative Colonoscopy

Recently, new indications for operative colonoscopy have been developed.

### 6.1. Endoscopic Therapy for Diverticular Abscesses

When the diverticulitis is accompanied by abscess formation, and if it is persistent after the administration of antibiotics, percutaneous drainage is the most frequently attempted intervention to treat the loco-regional inflammation. However, percutaneous drainage is technically not always easy and depends on the location and the abscess size. In particular, percutaneous abscess drainage poses a risk of injuring the intestine, vessels, or ureters when the abscess is located in the area surrounded by the small intestine or regions close to the vessels or ureters. A novel method of intraluminal abscess drainage and lavage using colonoscopy has been described by Kosugy et al. [59]. The candidate to be treated with this technique is the localized abscess caused by diverticulitis. Interestingly, bowel preparation is unnecessary, and a small amount of enema may be applied for preparation, together with mild sedation. After locating the diverticula responsible for causing the abscess using either CT or ultrasound, before performing the procedure under endoscopy, the diverticulum accountable for the abscess formation is punctured using either endoscopic forceps or a catheter, with purulent exudate coming from the puncture hole. Thus, the abscess cavity is investigated under fluoroscopy with a contrast agent to assess whether the cavity is localized and has not expanded to a wide area of the abdominal cavity. The abscess cavity is washed with saline solution (500 mL) through the inserted catheter until the draining water becomes completely clear. Although few cases have been treated with this technique, it seems safe to perform, with a failure rate of <15% [49].

### 6.2. Endoscopic Therapy for Colonic Strictures Associated with Diverticular Disease

Diverticular-associated colonic strictures are thought to occur due to recurrent episodes of acute diverticulitis causing fibrosis of a segment of the colon, most commonly the sigmoid colon. These diverticulitis episodes can be clinical or subclinical [60] and do not respond to endoscopic dilation, nor is endoscopic stricturoplasty usually therapeutically successful [61]. Therefore, a treatment option is the placement of self-expanding metal stents (SEMSs), which are generally used as a “bridge” to surgery. The SEMS is composed of a metal alloy and is advanced wire-guided across the strictured segment, possibly through fluoroscopy, where it is deployed. A SEMS is used only for symptomatic significant bowel obstruction, and there is no role for prophylactic SEMS placement. A CT examination should previously identify the area of interest; after appropriate lower bowel cleansing with enemas (tap water enemas usually suffice), the endoscopic instrument is advanced to the area of interest, and a stent is selected that is at least 4 cm longer than the area of interest (2 cm proximally and distally) [62,63]. The overall efficacy, defined as successful colon decompression, allowing for a complete colonic decompression and a one-stage surgical intervention, has been reported at 43% to 95% [64]. A recent review recommended against the routine use of SEMSs for the treatment of diverticular-associated colonic strictures. Still, they can be used in select patients with planned surgical intervention within one month [64]. These recommendations have been developed because SEMS may have early (<30 days, including perforation, stent failure after successful technical placement, stent migration re-obstruction, pain, and bleeding) and late (>30 days, including re-obstruction, tenesmus, incontinence, fistula, migration, and perforation) complications. In DD, the overall complication rate following SEMS for diverticular stricture is reported to be 6–43% [65].

### 6.3. Elective Endoscopic Clipping for the Eradication of Diverticula

Another exciting use of therapeutic endoscopy in managing DD is endoscopic clipping to eradicate the diverticula. This treatment aims to close the diverticula in patients with recurrent attacks of diverticular bleeding, preventing the disease’s recurrence. Haji et al., using a standard endoclip, closed the diverticular neck completely, eradicating most diverticula. Significantly, hardly any complications were recorded, and no diverticula recurrence was found at a one-year follow-up [66]. This is a promising technique, especially in patients with recurrent episodes of diverticular bleeding.

## 7. Conclusions

Colonoscopy plays an essential role in the management of DD. Of course, it requires advanced skills in recognizing situations of particular risk, such as AD with or without perforation signs, and in exceptional cases, such as massive diverticulosis with challenging-to-find colonic lumen, the presence of narrow angles, rigidly fixed lumen, or passing stenosis. The use of stiff thin-caliber endoscopes can be helpful in some cases; moreover, using a transparent cap, especially in training endoscopists, and using CO_2_ or water infusion instead of air insufflation may decrease the patient’s discomfort and help the endoscopist to conclude the colonic exploration [67,68]. Therefore, colonoscopy in DD should be performed under the following conditions: (1) in patients with LGIB suspected from diverticula, to be performed within 24 h and with bowel cleansing; (2) in persistently symptomatic patients following a recent AD to exclude other diseases (colorectal cancer or IBD in primis), to be performed after at least seven days of treatment and with only cleansing with enemas; (3) after the resolution of an uncomplicated AD if a high-quality examination of the colon has not been recently performed, to be performed ≥6 weeks after the acute episode; and (4) always after the resolution of an episode of complicated AD, to be performed ≥6 weeks after the acute episode. Moreover, for patients with their first endoscopic diagnosis of DD or SCAD, current endoscopic classifications must be mandatorily applied to have a predictive value on the outcome of the disease.

## Figures and Tables

**Figure 1 jpm-12-01713-f001:**
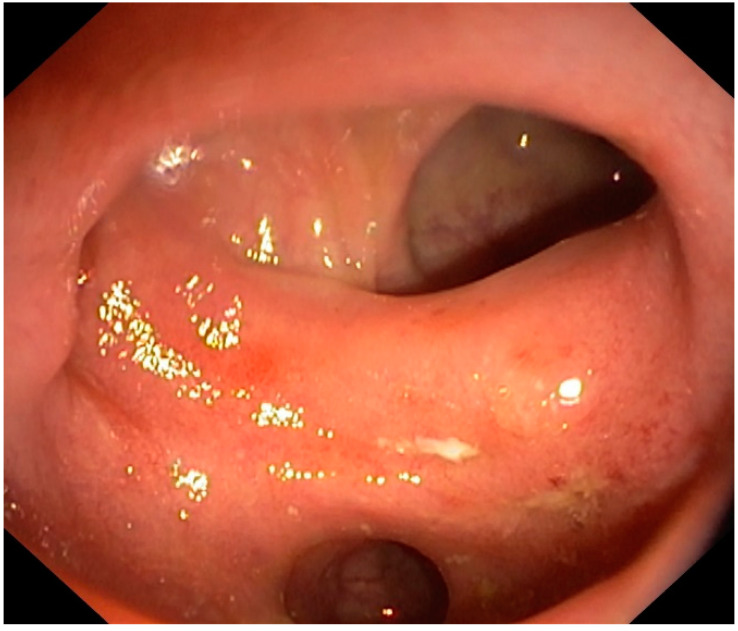
A 68-year-old female underwent a colonoscopy due to bloody diarrhea. Endoscopic appearance shows hyperemia and erosions on the colonic folds and the interdiverticular mucosa, with complete sparing of the diverticula. These lesions can be classified as type B SCAD.

**Figure 2 jpm-12-01713-f002:**
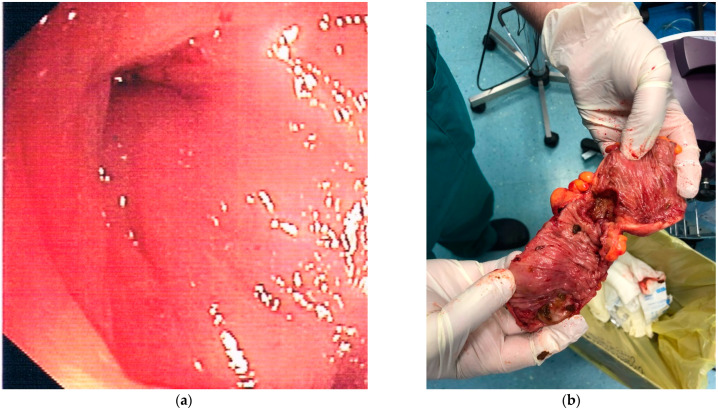
A 72-year-old inpatient female with acute diverticulitis complicated by stenosis underwent a colonoscopy due to persistent abdominal pain. A colonoscopy confirmed stenosis of the proximal descending colon (panel **a**), but histology was inconclusive. She underwent resection of the sigmoid and proximal descending colon (panel **b**), and histology was consistent with well-differentiated colonic adenocarcinoma.

**Figure 3 jpm-12-01713-f003:**
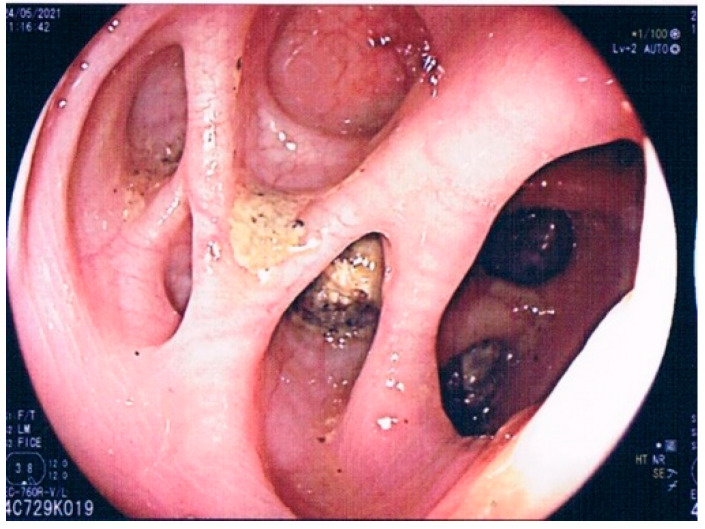
Massive diverticulosis with the stiffness of the lumen. Until the development of the DICA classification, the significance of these colonic characteristics was unknown. Now, this endoscopic picture can be scored as DICA 2 (left-sided diverticulosis: 2 points; >15 diverticula per district: 1 point; rigidity of the colonic wall: 4 points; total: 7 points), having an increased risk of developing complications related to the disease.

**Figure 4 jpm-12-01713-f004:**
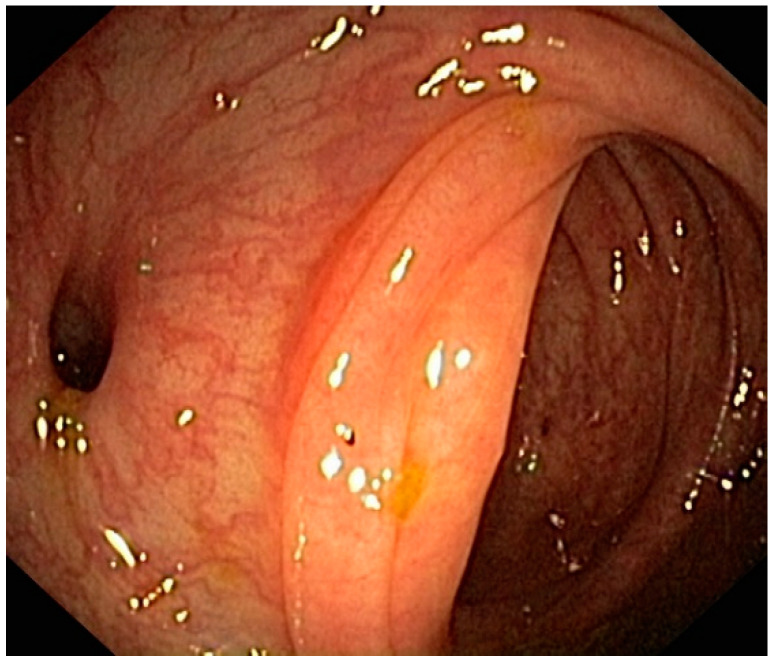
Endoscopic appearance of the colon eight weeks after an episode of uncomplicated diverticulitis. No signs of inflammation may be detected, and the patient risks being mis-scored according to the DICA classification.

**Figure 5 jpm-12-01713-f005:**
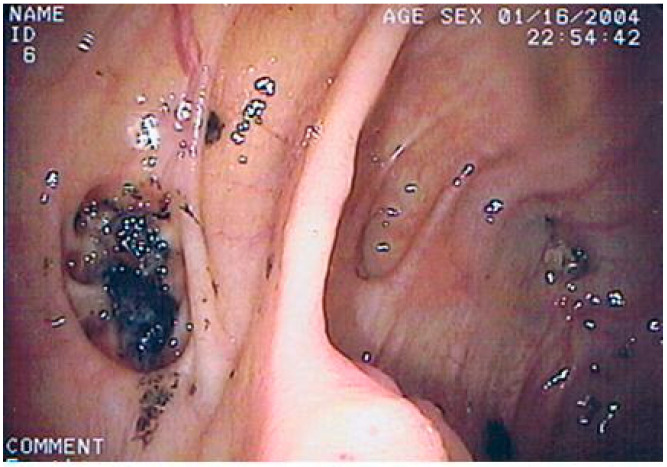
A 79-year-old man underwent colonoscopy due to massive rectal bleeding associated with severe anemia (hemoglobin 8.7 g/dL). The colonoscopy showed diverticulosis with signs of recent, but not current, bleeding.

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
