# Peer review of "When to Perform a Colonoscopy in Diverticular Disease and Why: A Personalized Approach"

_jpm, 2022, doi:10.3390/jpm12101713_

Round 1

Reviewer 1 Report

Very interesting work

Author Response

We thank the reviewer for his positive comment.

Reviewer 2 Report

It is a well-written narrative review. However, there are some concerns about this article. 1. The etiology of colonic diverticulum generally differs in western and oriental countries. The authors could discuss this issue. 2. The roles of diagnosis for colonic diverticula are different in modalities(e.g., barium enema, CTC). They could compare these modalities. 3. The authors should present literature search methods briefly. 4. Number of references is small for a full review article. 

Author Response

Reviewer 2.

It is a well-written narrative review. However, there are some concerns about this article.

We thank the reviewer for their helpful comments.

  1. The etiology of colonic diverticulum generally differs in western and oriental countries. The authors could discuss this issue.

Re; We reported in the text that localization and anatomical structure of colonic diverticula differ in Western and oriental countries. As suggested by the reviewer, we discussed the influence of genetics and diet on the location of the diverticula.

  1. The roles of diagnosis for colonic diverticula are different in modalities(e.g., barium enema, CTC). They could compare these modalities.

Re: We discuss the different modalities of diagnosis for colonic diverticula other than a colonoscopy and their indications.

  1. The authors should present literature search methods briefly.

Re: We reported the literature search methods of our narrative review.

  1. The number of references is small for a full review article. 

 Re; We added further references according to the reviewer’s suggestion.

Reviewer 3 Report

This is a well structured, well referenced and considered review of an important clinical topic. It brings together aspects of endoscopic diagnosis and management as they pertain directly to diverticulosis. For this reason it should be interesting to the readership. The aspect of personalisation of the approach is somewhat cursory however and more could be done to explain the individual patient characteristics that might lead to a greater or lesser need for colonoscopy. But the conclusions and summary of indications and timing for colonoscopy in the setting of diverticulosis are well thought out and sound. 

There are a few typos and very occasional errors of syntax so repeat proof reading should be applied. Where CO2 insufflation is mentioned as being superior to air, it seems to me a mention of the body of work regarding water insufflation should me made there too. The claim that unprepared colonoscopy should be avoided in diverticular bleeding due to lower completion rates and higher perforation rates should be backed up with references.

Author Response

Reviewer 3.

  1. This is a well structured, well referenced and considered review of an important clinical topic. It brings together aspects of endoscopic diagnosis and management as they pertain directly to diverticulosis. For this reason it should be interesting to the readership. The element of personalisation of the approach is somewhat cursory however and more could be done to explain the individual patient characteristics that might lead to a greater or lesser need for colonoscopy. But the conclusions and summary of indications and timing for colonoscopy in the setting of diverticulosis are well thought out and sound. 

Re: We thank the reviewer for his positive opinion of our review. We have tried to broaden the aspect of personalization of the approach by discussing in which cases CT colonography might be an alternative to colonoscopy.

  1. There are a few typos and very occasional errors of syntax so repeat proof reading should be applied.

Re: We repeat proofreading.

  1. Where CO2 insufflation is mentioned as being superior to air, it seems to me a mention of the body of work regarding water insufflation should be made there too.

Re: we thank the reviewer for this critical comment. We added in the text that current data suggest that water infusion decreases insertion pain and facilitate completion of difficult colonoscopy, as in diverticular disease.

  1. The claim that unprepared colonoscopy should be avoided in diverticular bleeding due to lower completion rates and higher perforation rates should be backed up with references.

Re: As  suggested by the reviewer, we added the following references regarding this claim:

Jensen DM, Machicado GA, Jutabha R, Kovacs TO. Urgent colonoscopy for the diagnosis and treatment of severe diverticular hemorrhage. N Engl J Med. 2000; 342:78–82.

Green BT, Rockey DC, Portwood G, Tarnasky PR, Guarisco S, Branch MS, Leung J, Jowell P. Urgent colonoscopy for evaluation and management of acute lower gastrointestinal hemorrhage: a randomized controlled trial. Am J Gastroenterol. 2005;100:2395–2402

Strate LL, Gralnek IM. ACG Clinical Guideline: Management of Patients With Acute Lower Gastrointestinal Bleeding. Am J Gastroenterol. 2016 Apr;111(4):459-74. doi: 10.1038/ajg.2016.41. Epub 2016 Mar 1. Erratum in: Am J Gastroenterol. 2016 May;111(5):755. PMID: 26925883; PMCID: PMC5099081.